# Longitudinal Associations of Adherence to the Dietary World Cancer Research Fund/American Institute for Cancer Research (WCRF/AICR) and Dutch Healthy Diet (DHD) Recommendations with Plasma Kynurenines in Colorectal Cancer Survivors after Treatment

**DOI:** 10.3390/nu14235151

**Published:** 2022-12-03

**Authors:** Daniëlle D. B. Holthuijsen, Martijn J. L. Bours, Eline H. van Roekel, Stéphanie O. Breukink, Maryska L. G. Janssen-Heijnen, Eric T. P. Keulen, Per M. Ueland, Øivind Midttun, Stefanie Brezina, Biljana Gigic, Andrea Gsur, Dieuwertje E. Kok, Jennifer Ose, Cornelia M. Ulrich, Matty P. Weijenberg, Simone J. P. M. Eussen

**Affiliations:** 1Department of Epidemiology, CARIM School for Cardivascular Diseases, Maastricht University, 6200 MD Maastricht, The Netherlands; 2Department of Epidemiology, GROW School for Oncology and Reproduction, Maastricht University, 6200 MD Maastricht, The Netherlands; 3Department of Surgery, GROW School for Oncology and Reproduction, NUTRIM School of Nutrition and Translational Research in Metabolism, Maastricht University Medical Centre+, 6200 MD Maastricht, The Netherlands; 4Department of Clinical Epidemiology, VieCuri Medical Centre, 5912 BL Venlo, The Netherlands; 5Department Internal Medicine and Gastroentology, Zuyderland Medical Centre Sittard-Geleen, 6162 BG Geleen, The Netherlands; 6Bevital AS, 5021 Bergen, Norway; 7Center for Cancer Research, Medical University of Vienna, 1090 Vienna, Austria; 8Department of General, Visceral and Transplantation Surgery, University of Heidelberg, 69120 Heidelberg, Germany; 9Division of Human Nutrition and Health, Wageningen University & Research, 6700 AA Wageningen, The Netherlands; 10Huntsman Cancer Institute, Salt Lake City, UT 84112, USA; 11Department of Population Health Sciences, University of Utah, Salt Lake City, UT 84108, USA; 12Department of Epidemiology, CAPHRI School for Care and Public Health Research Institute, Maastricht University, 6200 MD Maastricht, The Netherlands

**Keywords:** colorectal cancer survivorship, lifestyle, dietary patterns, dietary recommendations, kynurenines

## Abstract

The tryptophan-kynurenine pathway has been linked to cancer aetiology and survivorship, and diet potentially affects metabolites of this pathway, but evidence to date is scarce. Among 247 stage I-III CRC survivors, repeated measurements were performed at 6 weeks, 6 months, and 1 year post-treatment. Adherence to the World Cancer Research Fund/ American Institute for Cancer Research (WCRF) and Dutch Healthy Diet (DHD) recommendations was operationalized using seven-day dietary records. Plasma kynurenines of nine metabolites were analysed. Longitudinal associations of adherence to these dietary patterns and plasma kynurenines were analysed using confounder-adjusted linear mixed-models. In general, higher adherence to the dietary WCRF/AICR and DHD recommendations was associated with lower concentrations of kynurenines with pro-oxidative, pro-inflammatory, and neurotoxic properties (3-hydroxykynurenine (HK) and quinolinic acid (QA)), and higher concentrations of kynurenines with anti-oxidative, anti-inflammatory, and neuroprotective properties (kynurenic acid (KA) and picolinic acid (Pic)), but associations were weak and not statistically significant. Statistically significant positive associations between individual recommendations and kynurenines were observed for: nuts with kynurenic-acid-to-quinolinic-acid ratio (KA/QA); alcohol with KA/QA, KA, and xanthurenic acid (XA); red meat with XA; and cheese with XA. Statistically significant inverse associations were observed for: nuts with kynurenine-to-tryptophan ratio (KTR) and hydroxykynurenine ratio; alcohol with KTR; red meat with 3-hydroxyanthranilic-to-3-hydroxykynurenine ratio; ultra-processed foods with XA and KA/QA; and sweetened beverages with KA/QA. Our findings suggest that CRC survivors might benefit from adhering to the dietary WCRF and DHD recommendations in the first year after treatment, as higher adherence to these dietary patterns is generally, but weakly associated with more favourable concentrations of kynurenines and their ratios. These results need to be validated in other studies.

## 1. Introduction

Colorectal cancer (CRC) is the third most common type of cancer in both males and females, and the second most common cause of death worldwide [1]. In 2020 alone, there were nearly 1.9 million incident cases and roughly 900,000 deaths from CRC worldwide [1,2]. Ageing of the population and recent improvements in early detection and treatment of CRC have resulted in a marked rise in the number of CRC survivors [1,3,4]. CRC survivors often experience a lower health-related quality of life (HRQoL), even years after diagnosis [5,6]. It is, however, unclear which metabolic mechanisms can explain such lower HRQoL, and whether our diet may play a role in improving quality of life after cancer [7].

The tryptophan-kynurenine pathway, a catabolic process in which the essential amino acid tryptophan (Trp) is converted to kynurenine (Kyn) instead of serotonin, is recognized to be involved in HRQoL after cancer [8,9,10,11]. This pathway may play a role in inflammation [12], which is supported by the reported associations of kynurenines with chronic diseases and cancer [13,14]. Proinflammatory cytokines, such as interferon gamma, activate the enzyme indoleamine 2,3-dioxygenase (IDO1), thereby shifting Trp metabolism to Kyn production [8,12,15]. Kyn is then further metabolized into several other downstream products, such as 3-hydroxykynurenine (HK), kynurenic acid (KA), xanthurenic acid (XA), anthranilic acid (AA), 3-hydroxyanthranilic acid (HAA), picolinic acid (Pic), and quinolinic acid (QA), collectively referred to as kynurenines [8]. Some of these kynurenines are considered to have anti-oxidative, anti-inflammatory, and neuroprotective properties (i.e., KA, and Pic), others have pro-oxidative, pro-inflammatory, and neurotoxic properties (i.e., HK, and QA), while others have somewhat less well-characterized properties (i.e., XA, AA, and HAA) [8,16,17]. During inflammation, the balance between potentially neuroprotective and neurotoxic kynurenines is disturbed, which is reflected in their respective ratios. The kynurenine-to-tryptophan ratio (KTR) is a well-established marker of cellular immune activation [18,19]. In contrast, high levels of the KA/QA ratio, a ratio of two strong antagonists of N-methyl-D-aspartate (NMDA) receptors, has been considered to reflect neuroprotection [20].

Several lines of evidence have suggested a plausible link between diet and kynurenines. Firstly, dietary tryptophan, an essential amino acid and the substrate of the kynurenine pathway, and other downstream products (kynurenines) can be obtained through the diet [21,22,23]. Secondly, all enzymes in the pathway require minerals and vitamins; in particular, vitamin B2 and B6 play important roles as cofactors of these enzymes [8,24]. Only a few studies have examined the association between adherence to specific dietary patterns, in particular the Mediterranean diet, and kynurenines. A study in patients at high cardiovascular risk revealed that a higher adherence to the Mediterranean diet, a diet rich in vegetables, wholegrains, olive oil, nuts, and fish, resulted in higher concentrations of KA, and lower concentrations of HAA and QA [25]. Another study in healthy subjects found that higher adherence to the Mediterranean diet, Alternative Healthy Eating Index, and energy-adjusted Dietary Inflammation Index was associated with lower concentrations of plasma kynurenines [26]. 

Thus far, and despite accumulating evidence for a role of the kynurenine pathway in chronic diseases [13,14], the link between adherence to dietary patterns and kynurenines is understudied. Large population-based studies focusing on other dietary patterns, such as the World Cancer Research Fund and the American Institute for Cancer Research (WCRF/AICR) recommendations for cancer prevention [27], and national guidelines, such as the Dutch Healthy Diet (DHD) recommendations [28], in relation to kynurenines among CRC survivors are lacking. Hence, in this study, we aimed to assess the longitudinal associations of adherence to the dietary WCRF/AICR and DHD recommendations with metabolites of the kynurenine pathway in CRC survivors up to 12 months post-treatment.

## 2. Materials and Methods

### 2.1. Study Design and Population

Data were used from the Energy for Life after ColoRectal cancer (EnCoRe) study, an ongoing prospective, multicentre cohort study in CRC survivors in the Netherlands, initiated in 2012. Participants were recruited at diagnosis with stage I-III CRC at the Maastricht University Medical Center+, VieCuri Medical Center, and Zuyderland Medical Center. Participants were not eligible if they had been diagnosed with stage IV CRC, were younger than 18 years old, not residing in the Netherlands, non-Dutch speaking and reading, or had comorbidities that obstructed successful participation (e.g., Alzheimer’s disease). The study was approved by the Medical Ethics Committee of the University Hospital Maastricht and Maastricht University (Netherlands Trial Register no. NL6904) [29]. All participants signed written informed consent prior to participation.

### 2.2. Data Collection

Data collected until 1 November 2016 were used for the current analyses, because the metabolites of the kynurenine pathway have been assessed for participants followed-up until this date [30]. In accordance with standard operating procedures, trained dietitians collected data during home visits at 6 weeks (*n* = 247), 6 months (*n* = 199), and 12 months (*n* = 162) post-treatment (Figure 1). Response rates at follow-up measurements were >90%, and 14 participants had passed away until the datafreeze of November 2016. The declining number of participants at the subsequent time points is explained by the fact that not all participants included at diagnosis from April 2012 onwards had reached these time points in November 2016.

### 2.3. Dietary Intake

Participants completed a structured dietary record for seven consecutive days to obtain quantitative data on food and beverage intake. Consumed meals, food, and beverages with details on brand names, portion sizes, and preparation were reported in the dietary records. Participants received oral and written instructions on how to complete the dietary record. Upon receipt of the completed dietary records, dietitians checked them, and when necessary, they contacted the participants to clarify any incomplete and/or inconsistent entries. Nutrient intake per day was calculated using food calculation software (Compl-eat, Wageningen University, The Netherlands), based on the Dutch Food Composition Database (NEVO-2011). Two dietary scores, a score of adherence to the 2018 dietary WCRF/AICR recommendations [31,32,33] and a score of adherence to the 2015 DHD recommendations [34], were calculated according to methods previously published.

The score of adherence to the 2018 dietary WCRF/AICR recommendations was based on five recommendations regarding plant-based foods, ultra-processed foods, meat consumption, and sugary and alcoholic drinks. Briefly, the recommendations are to eat mostly plant-based foods by having a higher intake of fruit, vegetables, and dietary fibre; to limit the consumption of fast foods; to limit animal-based food products by reducing red and processed meat intake; to cut down on sugary drinks; and to limit the consumption of alcoholic drinks [27]. Each recommendation was given a score of 1 point (complete adherence), 0.5 point (partial adherence), and 0 points (non-adherence) to calculate the adherence score [31]. Predefined cut-off values of the dietary WCRF/AICR recommendations were used as described by Kenkhuis et al. [33], except for the recommendation on fast food where cut-offs were based on tertiles. The adherence score to the dietary WCRF/AICR recommendations ranged from 0 to 5, with a higher score reflecting better adherence (i.e., more healthy dietary habits).

The score of adherence to the 2015 DHD recommendations was based on fifteen recommendations for specific food groups, expressing the national guidelines for a healthy diet [28]. Briefly, the recommendations are to consume sufficient vegetables and fruit (≥200 g/d for both), whole-grain products (≥90 g/d), legumes (at least once weekly), unsalted nuts (≥15 g/d), fish (1 serving/wk); and tea (3 cups/d); to replace refined cereal products by whole-grain products, butter, hard margarines and cooking fats by soft margarines, liquid cooking fats and vegetable oil, and unfiltered coffee by filtered coffee; and to limit the use of red meat, particularly processed meat (quantity unspecified); sugar-containing beverages, alcohol (none or ≤1 glass/d), and salt (≤6 g/d) [28]. We did not operationalize the coffee and the salt component, as the use of filtered vs. unfiltered coffee was not specified and information on added salt intake was not collected, because this was deemed unreliable. Each recommendation was given a score between 0–10 points, depending on the extent of adherence to that particular recommendation. For the healthy food components, the minimum score (0) was given when there was no consumption of this component, and the maximum score (10) was given when intake was equal to the cut-off value or higher. For the unhealthy food components, 0 points were assigned if intake was above the threshold value, and 10 points were assigned if intake was equal to or lower than the cut-off value [34]. Subsequently, the adherence score to the DHD recommendations ranged from 0 to 130, with a higher score representing better adherence (i.e., more healthy dietary habits).

### 2.4. Blood Collection

EDTA plasma samples were collected, divided into aliquots, and stored at −80 °C until analysis. Samples were shipped on dry ice to the laboratory of Bevital AS in Bergen, Norway (www.bevital.no, accessed on 20 July 2022). Circulating concentrations of nine kynurenine metabolites including tryptophan (Trp), kynurenine (Kyn), 3-hydroxykynurenine (HK), kynurenic acid (KA), xanthurenic acid (XA), anthranilic acid (AA), 3-hydroxyanthranilic acid (HAA), picolinic acid (Pic), and quinolinic acid (QA) were analysed using liquid chromatography-tandem mass spectrometry (LC/MS-MS) [35]. Creatinine, an index of renal function, and neopterin, a marker for immune system activation, were analysed by LC/MS-MS as well [36]. Duplicate samples with known biomarker concentrations, as well as calibrator samples and a blank vial were added to each assay tray by Bevital to control for quality of the samples. The coefficients of within-day and between-day variation for tryptophan and kynurenines were 3.0–9.5% and 5.7–16.9%, respectively [35].

From the individual metabolites of the kynurenine pathway, we calculated relevant ratios. The kynurenine-to-tryptophan ratio (KTR), a well-established marker of inflammation, was calculated by dividing the plasma concentration of Kyn (in μmol/L) by the plasma concentration of Trp (in µmol/L), multiplied by 1000. The hydroxykynurenine ratio (HKr) was calculated by dividing the plasma concentration of HK (in nmol/L) by the sum of plasma concentrations of KA (in nmol/L), XA (in nmol/L), AA (in nmol/L) and HAA (in nmol/L). The 3-hydroxyanthranilic-acid-to-3-hydroxykynurenine ratio (HAA/HK), an indicator of intracellular functional status of pyridoxal 5′-phosphate (PLP) [37], was calculated by dividing the plasma concentration of HAA (in nmol/L) by the plasma concentration of HK (in nmol/L). The kynurenic-acid-to-quinolinic-acid ratio (KA/QA) was calculated by dividing the plasma concentration of KA (in nmol/L) by the plasma concentration of QA (in nmol/L).

### 2.5. Sociodemographic, Clinical and Lifestyle Variables

Age, sex, and clinical information (i.e., cancer stage, tumour location, and cancer treatment) were derived from medical records. Self-reported information was available on current smoking status (current, former, or never), on presence of stoma at all follow-up measurements, and on highest attained education level (categorized into high, medium, or low) only at diagnosis. The presence of comorbidities at all follow-up measurements was assessed using the Self-Administered Comorbidity Questionnaire [38]. Height (m) and body weight (kg) were measured during home visits to calculate body mass index (BMI) as weight/height² (kg/m²). Self-reported hours/week of light-intensity physical activity (LPA) and moderate-to-vigorous physical activity (MVPA) were determined at all follow-up measurements by the Short Questionnaire to Assess Health-enhancing physical activity (SQUASH) [39,40]. Total physical activity was calculated by adding the self-reported hours/week of LPA and MVPA. Total energy intake (kcal/week) was calculated using the seven-day dietary records.

### 2.6. Statistical Analyses

Population characteristics are presented as means and standard deviations (SD) for normally distributed variables, medians and interquartile ranges (IQR) for skewed variables, and frequencies (percentages) for categorical variables.

Linear mixed models were used to assess changes in plasma kynurenine concentrations over time, and to assess longitudinal associations of adherence to the dietary WCRF/AICR and DHD recommendations, and their individual dietary components with metabolites of the kynurenine pathway between 6 weeks and 12 months post-treatment as dependent variables. A random intercept for each subject was added to all models. The use of random slopes was tested with a likelihood-ratio test; when the model fit improved significantly, random slopes were added in the models.

A priori defined confounders were based on the literature [41] and their hypothesized relations with the exposure and outcome variables. All analyses were adjusted for sociodemographic factors, including age at enrolment (years), and sex (male, female); clinical factors, including creatinine (µmol/L) as an index of renal function, number of comorbidities (0, 1, ≥2), presence of stoma (yes, no), chemotherapy (yes, no) and time since end of treatment (weeks); and lifestyle-related factors, including BMI (kg/m²), total physical activity (hours/week), total energy intake (kcal/week), and smoking status (current, former, never). We further applied the 10% change-in-estimate method [42] for assessing an additional set of potential confounders, including education level (low, medium, high), and radiotherapy (yes, no); only education level led to a >10% change in β estimate in the model with age, sex, and renal function, and was, therefore, included in the fully adjusted model.

A hybrid model was used to disentangle two components: the inter-individual component and the intra-individual component. The inter-individual component reflects average differences between participants over time, obtained using centred person-mean values over all time points. The intra-individual component reflects average changes within participants over time, obtained using deviation scores of differences between observations at separate time points, and the person-mean value [43]. Both exposure and outcome variables were standardized using the average of the standard deviations across all follow-up measurements to allow direct comparisons of the regression coefficients regardless of the exposures and outcomes modelled. False discovery rate (FDR) adjustment of p-values using the Benjamini–Hochberg method was applied to adjust for multiple testing for each diet-kynurenine combination (*p*-values < 0.05 were considered significant) [44]. 

All analyses were conducted using Stata (version 17.0, Statacorp).

## 3. Results

### 3.1. Participant Characteristics

This prospective cohort study consisted of 325 participants at diagnosis (Figure 1), and 247 participants were included in the analyses (6 weeks to 12 months post-treatment) (Table 1). Of these 247 participants, 68.8% were male, and the mean age at 6 weeks post-treatment was 66.7 (SD: 9.1) years. Almost one third of the study population (32.4%) was diagnosed with stage I CRC, 23.5% with stage II CRC, and 44.1% with stage III CRC. Most participants were diagnosed with colon cancer (61.1%), while 38.9% of participants were diagnosed with rectum cancer. Almost all participants underwent surgery (89.5%), and 37.7% and 27.5% received chemotherapy and radiotherapy, respectively. Additional participant characteristics are summarized in Table 1.

### 3.2. Dietary Intake and Metabolites and Ratios of the Kynurenine Pathway

At 6 weeks post-treatment, the mean score of adherence to the dietary WCRF/AICR recommendations was 1.9 (SD: 0.7; range: 0.25–4.0) and remained relatively stable over time (Table 2). The mean score of adherence to the DHD recommendations was 60.7 (SD: 14.8; range: 26.9–96.9) at 6 weeks post-treatment and also appeared to remain relatively stable over time. Intake according to the individual components of the dietary adherence scores can be found in Table 2.

Plasma concentrations of metabolites of the kynurenine pathway and metabolite ratios are presented in Table 3. The mean Trp concentration was 65.7 (SD: 11.7) µmol/L at 6 weeks post-treatment and increased significantly over time up to 12 months post-treatment (*p* = 0.014). In contrast, mean Kyn concentration was 2.0 (SD: 0.5) µmol/L at 6 weeks post-treatment, and decreased significantly over time (*p* = 0.005). Plasma concentrations of the other kynurenine metabolites and relevant ratios at the follow-up time points are summarized in Table 3. KA (*p* < 0.001), Pic (*p* = 0.010), XA (*p* < 0.001), KA/QA ratio (*p* < 0.001), and HAA/HK ratio (*p* = 0.001) increased statistically significantly from 6 weeks post-treatment to 12 months post-treatment, while QA (*p* < 0.001) KTR (*p* < 0.001), and HKr (*p* < 0.001) decreased statistically significantly from 6 weeks post-treatment to 12 months post-treatment.

### 3.3. Longitudinal Associations of Adherence to Dietary WCRF/AICR and DHD Recommendations with Metabolites of the Kynurenine Pathway and Metabolite Ratios

Generally, we found no significant associations of adherence to the dietary WCRF/AICR and DHD recommendations with metabolites of the kynurenine pathway or ratios. However, data revealed a tendency for an association between higher adherence to the dietary WCRF/AICR and DHD recommendations with lower concentrations of kynurenines and ratios with pro-oxidative, pro-inflammatory, and neurotoxic properties (HK, QA, KTR, and HKr), and higher concentrations of kynurenines and ratios with anti-oxidative, anti-inflammatory, and neuroprotective properties (KA, Pic, KA/QA, and HAA/HK) (Figure 2 and Figure 3). Forest plots of kynurenines and ratios with the most striking results are displayed in Figure 2 and Figure 3, and forest plots and tables of all other kynurenines and ratios can be found in Appendix A. None of the associations remained statistically significant after adjustment for multiple testing (Appendix A).

Figure 2 and Figure 3 also show associations of individual components of the dietary scores with kynurenines and ratios (see also Appendix A). In line with associations observed for dietary scores, we observed a tendency that higher intake of healthy food components (vegetables, fruit, legumes, fibre, whole grain, nuts, lean and fatty fish, and tea) was associated with lower concentrations of HK, QA, KTR, and HKr, and with higher concentrations of KA, Pic, KA/QA and HAA/HK. Notably, the association of nuts with the KA/QA ratio (standardized β: 0.10; 95%CI: 0.04, 0.16), KTR (−0.10; −0.17, −0.03), and HKr (−0.09; −0.15, −0.04) remained statistically significant after adjustment for multiple testing (Appendix A).

Additionally, we observed a tendency for associations of higher intakes of unhealthy food components (soft and hard fats, refined grain, ultra-processed foods, red and processed meat, and sugar-sweetened drinks) with higher concentrations of HK, QA, KTR, and HKr and with lower concentrations of KA, Pic, KA/QA, and HAA/HK. The association of red meat and ultra-processed foods with XA (red meat: 0.11; 0.04, 0.18 and ultra-processed foods: −0.11; −0.17, −0.04), sweetened beverages and ultra-processed foods with KA/QA (sugar-sweetened beverages: −0.09; −0.16, −0.03 and ultra-processed foods: −0.11; −0.18, −0.04), and red meat with HAA/HK (−0.14; −0.22; −0.06) remained statistically significant after adjustment for multiple testing (Appendix A).

Interestingly, we found a tendency for an association between higher intake of alcohol with lower concentrations of HK, QA, KTR, and HKr, and with higher concentrations of KA, Pic, KA/QA, and HAA/HK. The associations with KA (0.12; 0.03, 0.20), XA (0.18; 0.09, 0.28), KTR (−0.25; −0.40, −0.11), and KA/QA (0.22; 0.12, 0.31) remained statistically significantly after adjustment for multiple testing (Appendix A). Excluding the alcohol component in the total dietary scores generally resulted in stronger overall associations. Similar associations were observed for cheese with kynurenines and ratios, though, only the association with XA (0.15; 0.07, 0.23) remained statistically significant after adjustment for multiple testing (Figure 3, Appendix A).

We observed no clear tendency in the association of adherence to the dietary WCRF/AICR or DHD recommendations with metabolites whose properties are less well characterized (Figure 2 and Figure 3). Although not part of the kynurenine pathway, associations of diet and neopterin (biomarker for interferon-gamma activity) showed essentially similar associations as were observed for KTR.

Notably, the significant overall associations were exclusively driven by the inter-individual component, indicating that a 1SD difference in intake between individuals over time was associated with differences in kynurenines or ratios, instead of changes over time within individuals. In particular, inter-individual associations were statistically significant for alcohol and ultra-processed foods with XA (alcohol: 0.20; 0.08, 0.31 and ultra-processed foods: −0.11; −0.17, −0.04), red meat with HAA/HK ratio (−0.17; −0.30, −0.04), and alcohol, ultra-processed foods and sugar-sweetened drinks with KA/QA (alcohol: 0.25; 0.14, 0.37 and ultra-processed foods: −0.14, −0.21, −0.07 and sugar-sweetened drinks: −0.21; −0.32, −0.10) (Figure 2 and Figure 3). Other significant inter-individual associations can be found in Appendix A.

## 4. Discussion

This study investigated longitudinal associations of adherence to the dietary World Cancer Research Fund/American Institute for Cancer Research and DHD recommendations with plasma concentrations of metabolites of the kynurenine pathway in CRC survivors up to 12 months post-treatment. This study revealed a tendency for higher adherence to the dietary WCRF/AICR and DHD recommendations to be associated with higher concentrations of kynurenines which have been reported to have anti-oxidative, anti-inflammatory, and neuroprotective properties (kynurenic acid (KA) and picolinic acid (Pic)), and with lower concentrations of kynurenines which have been reported to have anti-oxidative, anti-inflammatory, and neurotoxic properties (3-hydroxykynurenine (HK) and quinolinic acid (QA)) [17], but none of the associations were statistically significant. In line with this, higher intake of healthy food components tended to be associated with higher concentrations of neuroprotective kynurenines, and lower concentrations of neurotoxic kynurenines, whereas higher intake of unhealthy food components tended to be associated with lower concentrations of neuroprotective kynurenines, and higher concentrations of neurotoxic kynurenines. In addition, the observed associations of adherence to the WCRF and DHD recommendations with kynurenines appeared to be driven by differences between individuals over time, and not by changes within individuals.

The observed directions of the associations between adherence to the dietary WCRF/AIRC and DHD recommendations and kynurenines are generally in line with previous studies examining the association of adherence to the Mediterranean diet [25,26] and Alternative Healthy Eating Index [26] with metabolites of the kynurenine pathway in patients at high cardiovascular risk and healthy subjects. However, the observed tendency for a positive association between adherence to the dietary WCRF/AIRC recommendations and KA in the present study is contrary to previously reported inverse associations of adherence to the Mediterranean diet and Alternative Healthy Eating Index with KA in healthy subjects [25,26]. In addition, the observed tendency for a positive association between adherence to the dietary WCRF/AICR and DHD recommendations and HAA concentrations in the present study is also contradictory to that of Yu et al. [25] and Li et al. [26] who found inverse associations of adherence to the Mediterranean diet and Alternative Healthy Eating Index with HAA in healthy subjects at high cardiovascular risk, and healthy subjects, respectively. These rather contradictory results between studies may be due to different functions of HAA and KA in different health conditions [17]. Other explanations are differences in the assessment of dietary intake (Food Frequency Questionnaire (FFQ) versus seven-day dietary records), the smaller sample size, and different populations. It is important to mention that the observed associations in the present study were generally weak and often lacked statistical significance. Therefore, chance findings cannot be ruled out. Nevertheless, the overall consistency of results from the current study with former studies indicates that dietary patterns are potentially associated with metabolites of the kynurenine pathway.

The overall associations between adherence to dietary recommendations and metabolites of the kynurenine pathway were attenuated when the alcohol component was excluded from the total dietary scores. The observed null associations between dietary scores and metabolites of the kynurenine pathway may be explained by the striking associations we found for alcohol with kynurenines. Alcoholic beverages are considered an ‘unhealthy’ food component, as consistent evidence demonstrates that high alcohol consumption increases the risk of cancer and coronary heart diseases [28,45]. Therefore, we expected that higher consumption of alcoholic beverages would be associated with lower concentrations of neuroprotective kynurenines, and with higher concentrations of neurotoxic kynurenines, as demonstrated in previous studies in patients diagnosed with alcohol use disorder [46,47,48]. Remarkably, our data revealed the opposite, though not always statistically significant. This discrepancy may be explained by the differences in routes and dosages of ethanol administration. Intervention studies suggest that the activity of the rate-limiting enzyme of tryptophan degradation, tryptophan 2,3 dioxygenase (TDO), is enhanced by acute alcohol administration in humans, a single dose of 0.2 g/kg body weight ethanol, while observational studies suggest that TDO is inhibited after chronic alcohol consumption in humans [49,50]. It should be noted, however, that most studies on the effect of chronic alcohol consumption on Trp metabolism have been conducted exclusively in patients with alcohol use disorders, who cannot be compared to the colorectal cancer survivors in the present study. The seemingly beneficial effect of alcohol on kynurenines does not necessarily mean that alcohol is associated with more favourable concentrations of kynurenines. Instead, lower inflammation in the post-treatment period, reflected by a reduced KTR, may be associated with a better self-perceived quality of life, and those who start feeling better may be more likely to consume alcohol compared to participants that do not feel better yet [51]. 

To the best of our knowledge, literature on the associations of kynurenines with other food components than alcohol is lacking, except for fish intake and kynurenines [52]. In the present study, we did not observe a significant association between higher intake of lean fish and HAA concentrations, as was observed in cardiovascular patients [52]. However, the direction of the association was similar. In addition, our data revealed a significant positive association of cheese and red meat with XA, and of nuts with KA/QA, and a significant inverse association of nuts with KTR, and HKr, red meat with HAA/HK, ultra-processed foods with XA, and KA/QA, and sweetened beverages with KA/QA, but as far as we know, we are the first to observe this.

A major strength of the present study is its prospective design with repeated measurements of both dietary intake and plasma kynurenines. This study is the first to investigate associations between such comprehensive food components assessed by seven-day dietary records and a broad panel of metabolites of the kynurenine pathway. Moreover, there is no time-lag between the determinant and outcome, as the assessment of dietary intake and blood sampling occurred at one time. Response rates during follow-up were high (> 90%), and the number of missing data from intensive data collection was limited. Additionally, extensive data collection enabled us to adjust analyses for relevant confounders. On top of that, the linear mixed models enabled disentangling of inter- and intra-individual associations, thereby providing valuable insights into the nature of the longitudinal associations. Despite the strengths of this study, some limitations should be considered. The observational nature of our study limits the ability to draw conclusions regarding causality. In addition, a limited response rate at diagnosis (46%) to participate in the study might have resulted in selection bias and may affect the generalizability of the study. Furthermore, operationalization of dietary scores is challenging, and though we applied a priori selected evidence based dietary pattern scores [31,32,34], many choices could be considered. In addition, in the present study we classified kynurenines into anti-oxidative, anti-inflammatory, and neuroprotective, versus pro-oxidative, pro-inflammatory, and neurotoxic, versus less well-characterized properties to facilitate the presentation of these complex data. However, it should be noted that this was an arbitrary choice based on current evidence [17,53,54].

The current study is one of the first studies exploring the association between diet and plasma kynurenines, and these findings need to be validated by future studies that further elucidate the association between dietary intake and metabolites of the kynurenine pathway. Results with dietary scores always depend on how dietary scores are operationalized, and within a food component, there can be healthy and unhealthy food products, both of which may have different effects. Therefore, it is also worthwhile investigating the association between individual nutrients (macronutrients, vitamins, and minerals) and kynurenines. Another next step would be to investigate how diet-induced changes in kynurenines affect health-related outcomes in CRC survivors.

## 5. Conclusions

In conclusion, our findings suggest that CRC survivors might benefit from adhering to the dietary WCRF and DHD recommendations in the first year after treatment, as higher adherence to these dietary patterns is generally, but weakly associated with more favourable concentrations of kynurenines and their ratios. In addition, the findings of the present study suggest that a higher intake of healthy food components is associated with more favourable kynurenines and their ratios, whereas a higher intake of unhealthy food components is associated with less favourable kynurenines and their ratios. However, these results need to be validated in future studies.

## Figures and Tables

**Figure 1 nutrients-14-05151-f001:**
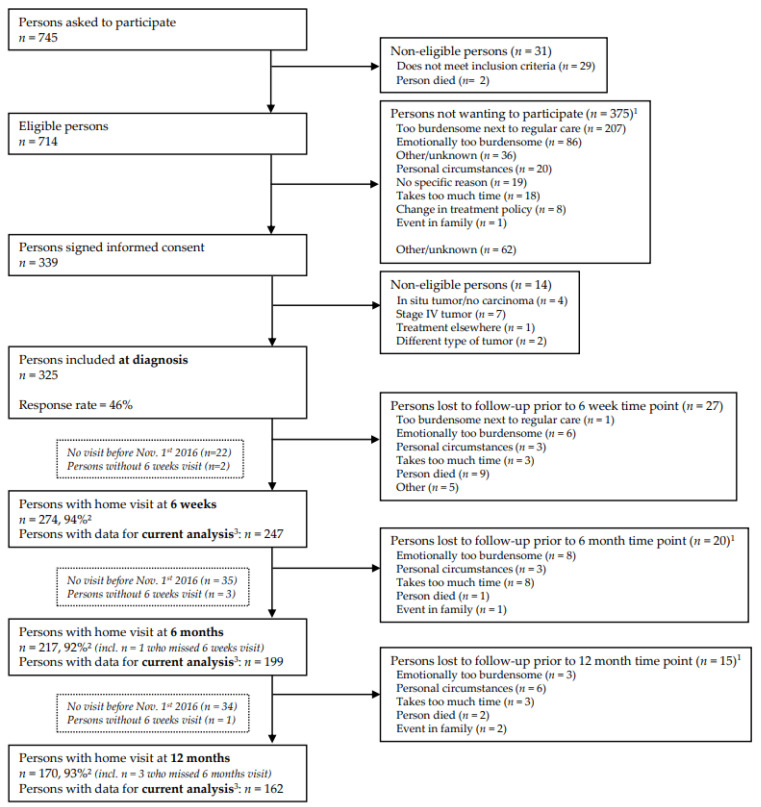
Flow diagram of the inclusion of participants within the EnCoRe study and the number of post-treatment measurements included in the analyses presented in this paper. Data of home visits performed before November 1st, 2016, were included in the analyses. ^1^ Totals do not add up because some individuals reported multiple reasons for non-participation. ^2^ Response rate = (persons with home visits)/(persons with home visits + persons lost to follow-up—persons died). The declining number of participants at the subsequent time points is because that not all participants included at diagnosis from April 2012 onwards had reached these time points in November 2016. ^3^ Since the current analysis was focused on dietary intake and kynurenines after colorectal cancer treatment, only post-treatment measurements with available data on dietary intake, kynurenines and covariates were included. The number of participants with available dietary intake and kynurenine data were, respectively, *n* = 256 and *n* = 251 at 6 weeks, *n* = 236 and *n* = 206 at 6 months, and *n* = 221 and *n* = 166 at 12 months post-treatment.

**Figure 2 nutrients-14-05151-f002:**
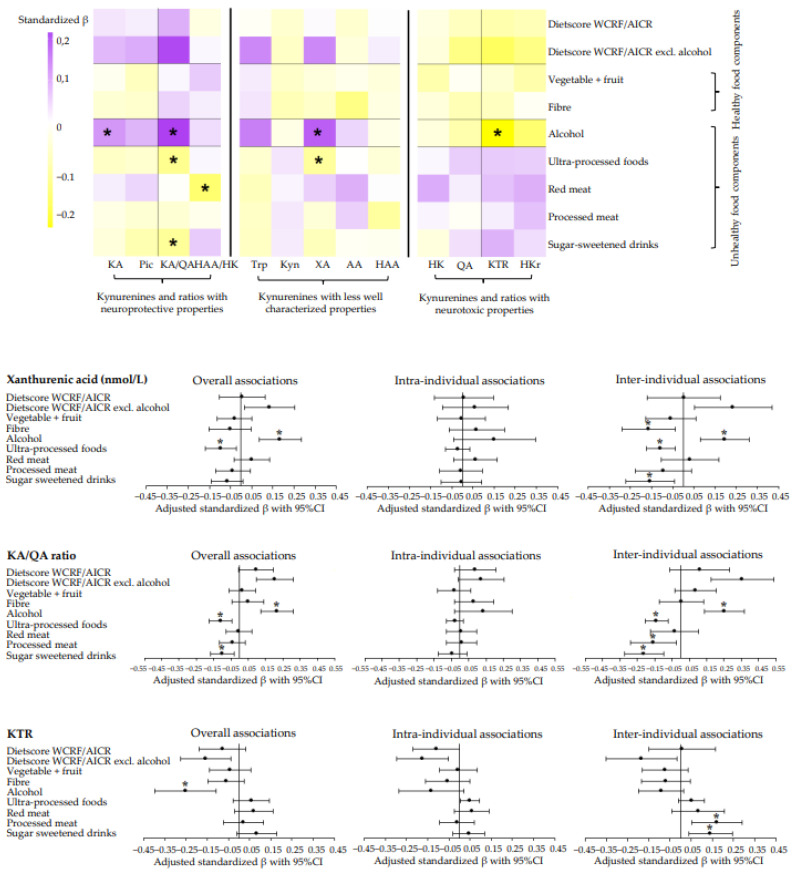
Heatmap and forest plots of confounder-adjusted linear mixed models between adherence to the dietary WCRF/AICR recommendations and metabolites and ratios of the kynurenine pathway in colorectal cancer survivors. Analyses were adjusted for age at enrolment (years), sex (male, female), renal function (µmol/L), time since end treatment (weeks), chemotherapy (yes, no), comorbidities (0, 1, ≥2), stoma (yes, no), educational level (high, medium, low), BMI (kg/m²), total PA (hours/week), smoking status (current, former, never) and energy intake (kcal/week). A random slope was added to the model for dietscore WCRF/AICR with HK, HAA/HK, KTR, and HKr; for dietscore WCRF/AICR excl. alcohol with Trp, HK, HAA/HK, KTR, and HKr; for vegetable and fruit with KA, AA, KTR, and HKr; for dietary fibre with HK, KA, QA, and HKr; for alcohol with Trp, HK, KA, QA, KTR, and HKr; for ultra-processed foods with QA, HAA/HK, KTR, and HKr; for red meat intake with HK, AA, and KTR; and for processed meat intake with AA, Pic, QA, HAA/HK, and KTR; and for sugar-sweetened drink intake with HAA/HK, KTR, and HKr. The standardized β-coefficient can be interpreted as the amount of SD difference in kynurenine concentration/ratio according to 1SD difference in dietary score/intake. * Statistically significant after FDR adjustment for multiple testing. Abbreviations: KA, kynurenic acid; Pic, picolinic acid; KA/QA ratio, kynurenic-acid-to-quinolinic-acid ratio; HAA/HK ratio, hydroxyanthranilic-to-hydroxykynurenine ratio; Trp, tryptophan; Kyn, kynurenine; XA, xanthurenic acid; AA, anthranilic acid, HAA, 3-hydroxyanthranilic acid; HK, 3-hydroxykynurenine; QA, quinolinic acid; KTR, kynurenine-to-tryptophan ratio; HKr, hydroxykynurenine ratio.

**Figure 3 nutrients-14-05151-f003:**
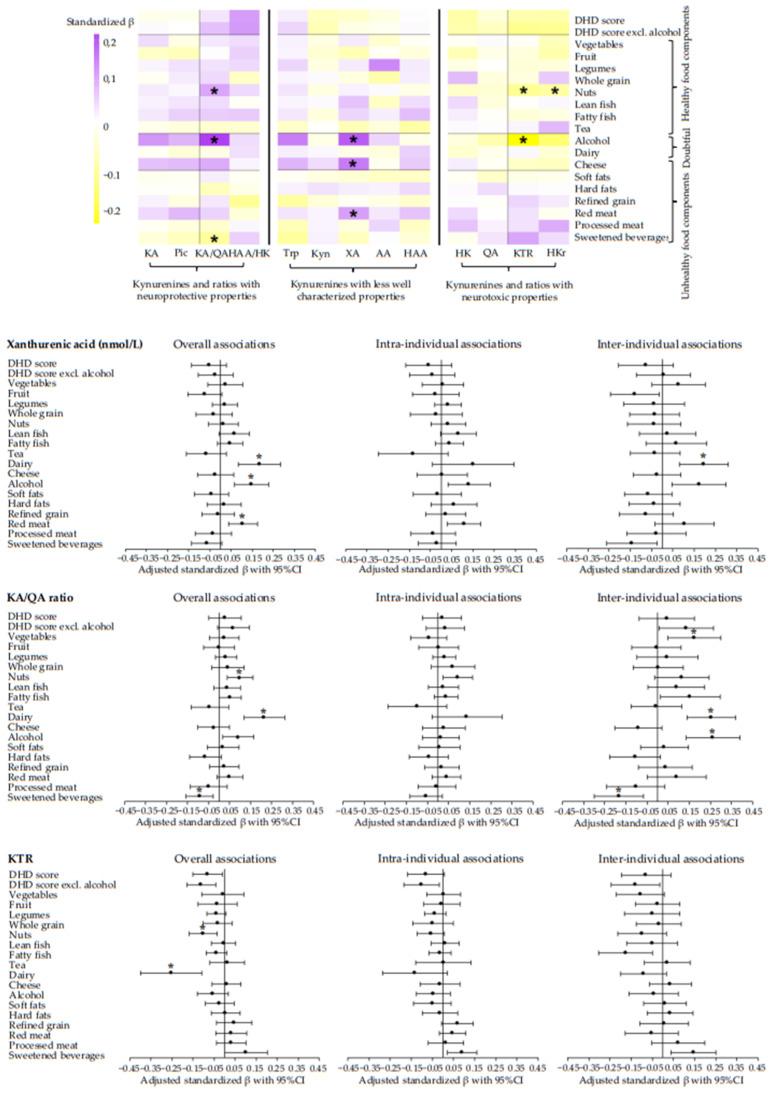
Heatmap and forest plots of confounder-adjusted linear mixed models between adherence to the DHD recommendations and metabolites and ratios of the kynurenine pathway in colorectal cancer survivors. Analyses were adjusted for age at enrolment (years), sex (male, female), renal function (µmol/L), time since end treatment (weeks), chemotherapy (yes, no), comorbidities (0, 1, ≥2), stoma (yes, no), educational level (high, medium, low), BMI (kg/m²), total PA (hours/week), smoking status (current, former, never), and energy intake (kcal/week). A random slope was added to the model for DHD score with XA; for vegetables with HK, KA, XA, Pic, and KTR; for fruit with HK, AA, Pic, KTR, and HKr; for legumes with AA; for whole grain with HK, AA, QA, KTR, and HKr; for nuts with Trp, HAA, HAA/HK, KTR, and HKr,; for lean fish with AA, QA, and HKr; for fatty fish with HAA, and HAA/HK; for tea with AA, and HKr; for alcohol with Trp, HK, KA, QA. KTR, and HKr; for dairy with HK, XA, and HKr; for cheese with Trp, QA, and KTR; for soft fats with HK, AA, and HKr; for refined grain with AA, QA, HAA/HK, KTR, and HKr; for red meat with Trp, QA, HAA/HK, and KTR; for processed meat with Trp, KA, AA, Pic, KA/QA, and HAA/HK; and for sweetened beverages and fruit juices with HAA/HK, and KTR. The standardized β-coefficient can be interpreted as the amount of SD difference in kynurenine concentration/ratio according to 1SD difference in dietary score/intake. * Statistically significant after FDR adjustment for multiple testing. Abbreviations: KA, kynurenic acid; Pic, picolinic acid; KA/QA ratio, kynurenic-acid-to-quinolinic-acid ratio; HAA/HK ratio, hydroxyanthranilic-to-hydroxykynurenine ratio; Trp, tryptophan; Kyn, kynurenine; XA, xanthurenic acid; AA, anthranilic acid, HAA, 3-hydroxyanthranilic acid; HK, 3-hydroxykynurenine; QA, quinolinic acid; KTR, kynurenine-to-tryptophan ratio; HKr, hydroxykynurenine ratio.

**Table 1 nutrients-14-05151-t001:** Sociodemographic, lifestyle, and clinical characteristics of participants included in the Energy for Life after ColoRectal cancer (EnCoRe) study at 6 weeks post-treatment.

	Total, *n* = 247 *
Sex (male), *n* (%)	170 (68.8)
Age (years), mean (SD)	66.7 (9.1)
Cancer location, *n* (%)	
Colon	151 (61.1)
Rectum	96 (38.9)
Cancer stage, *n* (%)	
Stage I	80 (32.4)
Stage II	58 (23.5)
Stage III	109 (44.1)
Cancer treatment, *n* (%)	
Surgery (yes)	221 (89.5)
Chemotherapy (yes)	93 (37.7)
Radiotherapy (yes)	68 (27.5)
Number of comorbidities, *n* (%)	
0 comorbidities	50 (20.2)
1 comorbidity	60 (24.3)
≥2 comorbidities	137 (55.5)
Stoma (yes), *n* (%)	78 (31.7)
BMI (kg m²), mean (SD)	27.8 (4.4)
Educational level, *n* (%)	
Low	62 (25.2)
Medium	98 (39.8)
High	86 (35.0)
Smoking status, *n* (%)	
Never	83 (33.9)
Former	140 (57.1)
Current	22 (8.9)
Physical activity, median (IQR)	
LPA (hours/week)	7.0 (12.5)
MVPA (hours/week)	7.2 (11.7)
Total PA (hours/week)	18.3 (21.0)
Total energy intake (kcal/week), mean (SD)	14,687.0 (3633.3)
Creatinine (µmol/L), mean (SD)	83.6 (18.8)
Neopterin (nmol/L), median (IQR)	15.9 (10.7)

Abbreviations: BMI, body mass index; LPA, light-intensity physical activity; MVPA, moderate-to-vigorous physical activity; PA, physical activity; SD, standard deviation; IQR, interquartile range. * Percentages may not add up to 100 due to rounding.

**Table 2 nutrients-14-05151-t002:** Dietary components of the WCRF/AICR and DHD recommendations of participants included in the Energy for Life after ColoRectal cancer (EnCoRe) study at 6 weeks, 6 months, and 12 months post-treatment.

	6 Weeks Post-Treatment (*n* = 247)	6 Months Post-Treatment (*n* = 199)	12 Months Post-Treatment (*n* = 162)
**WCRF/AICR diet score**	1.9 (0.7)	2.1 (0.7)	2.0 (0.7)
Fruit, vegetable and fibre intake			
Fruit intake (g/day)	119.1 (86.8)	123.3 (93.0)	108.8 (90.5)
Vegetable intake (g/day)	133.7 (71.8)	135.4 (73.5)	130.8 (69.2)
Dietary fibre intake (g/day)	21.0 (5.9)	21.0 (6.4)	21.1 (6.1)
Alcohol intake (g/day)	13.6 (18.6)	13.0 (19.2)	14.5 (19.3)
Sugar-sweetened drinks (g/day)	135.8 (147.4)	107.0 (134.9)	101.2 (114.9)
UPF (% energy)	35.1 (10.9)	33.2 (10.3)	33.3 (9.8)
Meat intake			
Red meat intake (g/week)	614.0 (311.5)	600.0 (278.8)	611.8 (312.1)
Processed meat intake (g/week)	326.7 (209.2)	312.5 (190.1)	323.8 (228.5)
**DHD total score**	60.7 (14.8)	62.2 (13.7)	61.3 (15.4)
Fruit intake (g/day)	117.5 (86.9)	122.2 (92.8)	107.0 (89.7)
Vegetable intake (g/day)	135.2 (71.6)	136.7 (72.4)	131.2 (70.2)
Legumes intake (g/day)	2.1 (8.0)	5.0 (15.0)	2.6 (8.2)
Nuts intake (g/day)	3.0 (7.2)	3.6 (7.8)	3.0 (7.2)
Grain intake			
Wholegrain intake (g/day)	113.6 (63.3)	120.6 (64.4)	122.8 (62.0)
Refined grain intake (g/day)	80.8 (52.2)	71.0 (48.1)	77.0 (51.3)
Alcohol intake (g/day)	13.6 (18.6)	13.0 (19.2)	14.5 (19.3)
Dairy intake			
Dairy intake (g/day)	187.1 (160.2)	165.9 (139.8)	157.0 (128.3)
Cheese intake (g/day)	28.4 (20.6)	28.5 (22.2)	28.0 (18.8)
Fish intake			
Fatty fish intake (g/day)	11.0 (20.4)	9.3 (14.1)	7.9 (11.6)
Lean fish intake (g/day)	11.5 (14.7)	9.8 (15.6)	10.6 (18.2)
Tea intake (g/day)	209.8 (255.3)	209.4 (270.4)	216.0 (265.2)
Fat intake			
Hard fats intake (g/day)	8.3 (11.1)	9.6 (12.7)	8.9 (12.3)
Soft fats intake (g/day)	22.1 (14.9)	19.2 (13.5)	21.6 (13.9)
Meat intake			
Red meat intake (g/day)	35.2 (26.4)	37.5 (27.0)	35.7 (27.3)
Processed meat intake (g/day)	49.6 (32.7)	48.0 (29.0)	50.8 (36.0)
Sweetened beverages and fruit juices (g/day)	106.8 (134.2)	79.4 (112.2)	72.4 (97.3)

Abbreviations: WCRF/AICR, World Cancer Research Fund/American Institute for Cancer Research; DHD, Dutch Healthy Diet; UPF, ultra-processed foods. All values are presented as means and standard deviations (SD).

**Table 3 nutrients-14-05151-t003:** Plasma concentrations of metabolites of the kynurenine pathway and metabolite ratios of participants included in the Energy for Life after ColoRectal cancer (EnCoRe) study at 6 weeks, 6 months, and 12 months post-treatment.

	6 Weeks Post-Treatment (*n* = 247)	6 Months Post-Treatment (*n* = 199)	12 Months Post-Treatment (*n* = 162)	*p*-Value
Tryptophan (µmol/L)	65.7 (11.7)	67.4 (13.1)	68.1 (11.8)	0.014 *
Kynurenine (µmol/L)	2.0 (0.5)	1.9 (0.5)	1.9 (0.4)	0.005 *
3-Hydroxykynurenine (nmol/L) ^1^	58.2 (31.2)	53.7 (26.3)	51.5 (24.8)	0.050
Kynurenic acid (nmol/L)	57.9 (26.5)	63.9 (28.8)	67.1 (32.9)	<0.001 *
Xanthurenic acid (nmol/L)	14.4 (8.1)	15.9 (7.5)	16.9 (8.6)	<0.001 *
Anthranilic acid (nmol/L) ^1^	16.8 (5.9)	17.5 (5.9)	17.9 (8.4)	0.084
3-Hydroxyanthranilic acid (nmol/L) ^1^	44.8 (14.4)	43.9 (14.9)	44.2 (14.5)	0.321
Picolinic acid (nmol/L)	36.7 (15.6)	37.5 (13.7)	38.8 (15.8)	0.010 *
Quinolinic acid (nmol/L)	628.6 (389.9)	609.8 (434.2)	539.2 (274.5)	<0.001 *
KTR	31.2 (10.1)	29.8 (11.3)	28.4 (8.2)	<0.001 *
HKr	0.45 (0.23)	0.39 (0.14)	0.36 (0.13)	<0.001 *
HAA/HK ratio	0.88 (0.33)	0.93 (0.48)	0.95 (0.35)	0.001 *
KA/QA ratio	0.11 (0.05)	0.12 (0.05)	0.14 (0.05)	<0.001 *

^1^ 6 weeks post-treatment: *n* = 245, 6 months post-treatment *n* = 198, 12 months post-treatment *n* = 162. Abbreviations: KTR, kynurenine-to-tryptophan ratio; HKr, hydroxykynurenine ratio; HAA/HK, 3-hydroxyanthranilic-acid-to-3-hydroxykynurenine ratio; KA/QA, kynurenic-acid-to-quinolinic-acid ratio. All values are presented as means and standard deviations (SD). * Indicates a statistically significant difference (*p* ≤ 0.05) between follow-up time points based on linear mixed models.

## Data Availability

Data described in the manuscript, code book, and analytic code will be made available upon request pending, e.g., application and approval, payment, or other. Requests for data of the EnCoRe study can be sent to Martijn Bours, Department of Epidemiology, GROW School for Oncology and Developmental Biology, Maastricht University, the Netherlands (email: m.bours@maastrichtuniversity.nl).

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
