# Peer review of "Longitudinal Associations of Adherence to the Dietary World Cancer Research Fund/American Institute for Cancer Research (WCRF/AICR) and Dutch Healthy Diet (DHD) Recommendations with Plasma Kynurenines in Colorectal Cancer Survivors after Treatment"

_nutrients, 2022, doi:10.3390/nu14235151_

Round 1
Reviewer 1 Report
This study discussed the associations between plasma kynurenines and colorectal cancer survivors after treatment. From the methodological perspective of this paper, the novelty and scope meet the standard of this journal.
Here are some of my personal recommendations. If there is a misunderstanding of what this paper does, feel free to clarify.
The authors chose to use the data from 2012 to 2016, which is a long time ago. What is the particularity of this group of data and can authors explain what are the reasons for this choosing.
The conclusion did not clearly describe the work of this article, but said some correct and useless words, such as “Healthy food is good for health’. The authors should learn the standard writing of the conclusion. Several key discussions in 4. Discussion should be included in the conclusion.
Author Response
Response to Reviewer 1 Comments
Referee 1
Point 1: The authors chose to use the data from 2012 to 2016, which is a long time ago. What is the particularity of this group of data and can authors explain what are the reasons for this choosing.
Response 1: We thank the referee for this question. We included participants with data up to the 1st of November 2016, because metabolites of the kynurenine pathway were determined only for these participants as part of the FOCUS consortium. We mentioned this in the method section (line 129-131). In reference 30, more information on this can be obtained.
Point 2: The conclusion did not clearly describe the work of this article, but said some correct and useless words, such as “Healthy food is good for health’. The authors should learn the standard writing of the conclusion. Several key discussions in 4. Discussion should be included in the conclusion.
Response 2: We thank the referee for the critical view on the conclusion. Although we appreciate the comment made, we are not sure if we understand it correctly. In the discussion (part 4), we compared our study findings with those of other studies, and we discussed methodological considerations. In the conclusion (part 5), we carefully formulated a “take-home message”, without too much repetition of the methodological considerations outlined in our discussion. In addition, we clearly state that our findings should be interpreted with caution and that these findings should be validated in future studies. We have discussed the wording of the conclusion/take-home message from both the abstract and the general discussion again with our team. Based on this, we have changed the wording slightly in the abstract, as well as the conclusion in part 5, to better align the conclusions in these sections.
Abstract: “Our findings suggest that CRC survivors might benefit from adhering to the WCRF and DHD recommendations in the first year after treatment, as higher adherence to these dietary patterns is generally, but weakly associated with more favourable kynurenines and their ratios. These results need to be validated in other studies.” (Abstract, line 56-59)
Conclusion part 5: “In conclusion, our findings suggest that CRC survivors might benefit from adhering to the WCRF and DHD recommendations in the first year after treatment, as higher adherence to these dietary patterns is generally, but weakly associated with more favourable kynurenines and their ratios. In addition, the findings of the present study suggest that higher intake of healthy food components is associated with more favourable kynurenines and their ratios, whereas higher intake of unhealthy food components is associated with less favourable kynurenines and their ratios. However, these results need to be validated in future studies.” (Conclusion, line 529-537)
We believe that we have carefully weighed the wording of our final conclusion/take-home message, and we feel that overall the conclusion is a valid reflection of the current study, as we are not over-interpreting our study findings. If the reviewer disagrees, we are willing to revise the conclusion, but we kindly ask the referee for further clarification.

Reviewer 2 Report
Dear, no comments - satisfied.
Author Response
Response to Reviewer 2 Comments
Point 1: Dear, no comments – satisfied.
Response 1: We thank the reviewer for the compliments. We are glad to read that this reviewer was satisfied with our work and had no further feedback to improve this paper.
